# Temperature and quantum anharmonic lattice effects on stability and superconductivity in lutetium trihydride

Roman Lucrezi [1], Pedro P. Ferreira [1,2], Markus Aichhorn [1] & Christoph Heil [1] ✉

In this work, we resolve conflicting experimental and theoretical findings related to the dynamical stability and superconducting properties of $Fm\bar{3}m$-LuH$_3$, which was recently suggested as the parent phase harboring room-temperature superconductivity at near-ambient pressures. Including temperature and quantum anharmonic lattice effects in our calculations, we demonstrate that the theoretically predicted structural instability of the $Fm\bar{3}m$ phase near ambient pressures is suppressed for temperatures above 200 K. We provide a $p-T$ phase diagram for stability up to pressures of 6 GPa, where the required temperature for stability is reduced to $T > 80$ K. We also determine the superconducting critical temperature $T_c$ of $Fm\bar{3}m$-LuH$_3$ within the Migdal-Eliashberg formalism, using temperature- and quantum-anharmonically-corrected phonon dispersions, finding that the expected $T_c$ for electron-phonon mediated superconductivity is in the range of 50–60 K, i.e., well below the temperatures required to stabilize the lattice. When considering moderate doping based on rigidly shifting the Fermi level, $T_c$ decreases for both hole and electron doping. Our results thus provide evidence that any observed room-temperature superconductivity in pure or doped $Fm\bar{3}m$-LuH$_3$, if confirmed, cannot be explained by a conventional electron-phonon mediated pairing mechanism.

The scientific community's interest in lutetium hydrides has been reignited overnight following Dasenbrock-Gammon et al. recent announcement of a room-temperature superconductor (RTS) at almost ambient pressure in the Lu-N-H system[1]. If this result would be confirmed, it could mark the discovery of a new, revolutionary class of lowest-pressure, highest-$T_c$ hydrides, paving the way for a plethora of technological advancements in fields ranging from medicine to quantum computing[2–9].

Although the proposed RTS phase has been characterized using several different experimental techniques in ref. 1, including X-ray diffraction (XRD), energy-dispersive X-ray, and Raman spectroscopy, the structural details of the *red matter* superconductor, as named by the authors due to the bright red color of the samples, remains unknown given the inability of conventional spectroscopy methods in measuring defect densities and fractional occupations of light elements such as H or N.

The main phase reported in ref. 1 to be superconducting is suggested to be LuH$_{3-\delta}$N$_\epsilon$ with $Fm\bar{3}m$ face-centered cubic (fcc) space-group (SG), based on the obtained diffraction patterns and comparisons with known lutetium hydrides[10–13]. A minor, non-superconducting phase B was also identified in all the obtained samples and assigned to a $Fm\bar{3}m$-LuH$_2$ structure[1].

Since the initial report by Dasenbrock-Gammon et al.[1], several experimental groups have invested significant efforts into replicating

[1]Institute of Theoretical and Computational Physics, Graz University of Technology, NAWI Graz 8010 Graz, Austria. [2]Universidade de São Paulo, Escola de Engenharia de Lorena, DEMAR, 12612-550 Lorena, Brazil. ✉e-mail: christoph.heil@tugraz.at

the experimental findings and gaining a better understanding of the proposed superconducting phase[14–23]. A few theoretical studies have also been made available[24–29], identifying the most likely phases that match the experimental data and investigating the possibility of achieving RTS in the Lu-N-H system within the electron-phonon (el-ph) coupling mechanism.

Independent crystal structure prediction studies have demonstrated that both the $LuH_2$ and $LuH_3$ compositions are thermodynamically stable in the Lu-N-H ternary phase diagram[24–26,28]. While these theoretical works have confirmed the presence of the $Fm\bar{3}m$-$LuH_2$ phase on the convex hull, they have also revealed that for $LuH_3$ the cubic $Fm\bar{3}m$ phase is not the thermodynamically most stable one close to ambient pressure[24–26,28]. In addition, phonon calculations for $Fm\bar{3}m$-$LuH_3$ within the commonly employed harmonic approximation suggest that this phase is dynamically highly unstable due to the presence of several imaginary modes in the vibrational spectra present in the whole extent of the Brillouin zone (BZ)[24,25,29].

The two $Fm\bar{3}m$ phases of $LuH_2$ and $LuH_3$, even though the latter is supposedly dynamically unstable, are of significant importance. On the one hand, recent studies argue that the actual composition observed in Dasenbrock-Gammon et al.'s work[1] is, in fact, N-doped or pure fcc $LuH_2$[14,16–18,20]. The differentiation here is quite tricky, as $LuH_2$ and $LuH_3$ share the same structural framework with H atoms occupying only tetrahedral sites or tetrahedral + octahedral sites, respectively. Moreover, their crystal cell volumes at ambient pressure are also quite similar ($V^0_{LuH_2} = 31.6$ Å vs. $V^0_{LuH_3} = 31.3$ Å) and they both match the reported XRD patterns of Dasenbrock-Gammon et al. for the main phase[24,25].

On the other hand, it has been predicted recently that $Fm\bar{3}m$-$LuH_2$ has a negligibly low $T_c$[24], while $Fm\bar{3}m$-$LuH_3$ is predicted to have finite critical temperatures smaller than 35 K[24]. It is important to note, however, that the calculations for the latter have been performed by neglecting all contributions of the imaginary modes to the el-ph coupling and, therefore, only serve as a rough estimate.

To get an accurate description of the physical properties in these systems, several effects have to be taken into account: (i) As has been demonstrated for various materials, anharmonicity and the quantum motion of ions, effects that are neglected in harmonic phonon calculations within the Born–Oppenheimer approximation, can have significant implications for certain materials[30,31,31–35]. These corrections can become particularly important for hydrogen-rich compounds[35–40], where the zero-point motion of H atoms can even modify the ground state structure, and anharmonic effects may significantly change the energy of high-frequency H vibrations. (ii) Given that the experiments for the Lu-N-H system have been conducted at elevated temperatures up to room temperature, including temperature effects may be crucial for an accurate modeling of the system as well. (iii) Furthermore,

doping with N, as suggested in ref. 1, could also impact the superconducting properties.

With this work, we want to clarify the conflicting experimental and theoretical findings related to the stability of $Fm\bar{3}m$-$LuH_3$ and provide an accurate description of its superconducting properties to understand if RTS at near-ambient pressure is attainable in this phase. Compellingly, by the inclusion of quantum ionic effects and anharmonicity within the framework of the stochastic self-consistent harmonic approximation (SSCHA)[41], we show that the $Fm\bar{3}m$-$LuH_3$ phase can be stabilized at temperatures above 200 K for pressures of 1 GPa and we also provide a $p–T$ phase diagram for stability up to 6 GPa. Moreover, by utilizing the Migdal-Eliashberg formalism[42,43], we predict a $T_c$ of approx. 50–60 K at a pressure of 2.8 GPa, and also probe the effects of small doping on $T_c$, providing evidence that any observed room-temperature superconductivity in pure or moderately doped $Fm\bar{3}m$-$LuH_3$, if present at all, cannot be explained by a conventional el-ph pairing mechanism.

## Results
### Temperature and quantum anharmonic lattice effects on stability

As mentioned above, the phonon dispersion of $Fm\bar{3}m$-$LuH_3$ (hereafter, we omit the $Fm\bar{3}m$ notation when referring to $Fm\bar{3}m$-$LuH_2$ or $Fm\bar{3}m$-$LuH_3$) at ambient pressures within the harmonic approximation exhibits several imaginary phonon modes throughout the whole BZ, shown in Fig. 1a as dashed black lines. This could either indicate a genuine structural instability of the phase or an inadequate theoretical description of the physical system. To resolve this issue, we have calculated the vibrational properties incorporating quantum ionic, anharmonic, and temperature effects on the same footing, as implemented in SSCHA[41]. The self-consistent harmonic approximation method effectively maps the full anharmonic and interacting lattice onto an auxiliary harmonic system. The actual system's ground state is then obtained by minimizing the free energy of the auxiliary harmonic system[44–47]. This minimization is performed stochastically via Monte Carlo summation and sampling over several consecutive ensembles arbitrarily populated by many structures. Within the density-functional theory (DFT) framework, each structure denotes a supercell with displaced atomic positions, where the supercell dimension determines the density of phonon wave vectors in the BZ[37]. Further information is available in the Method section, Supplementary Method 1, and refs. 32,36,39,41,48–50.

Noteworthy, the potential energy surface changes upon including quantum anharmonic lattice effects. For the high-symmetry $LuH_3$, the only structural degree of freedom is the lattice constant (synonymously, the unit cell volume), as all other structural parameters are

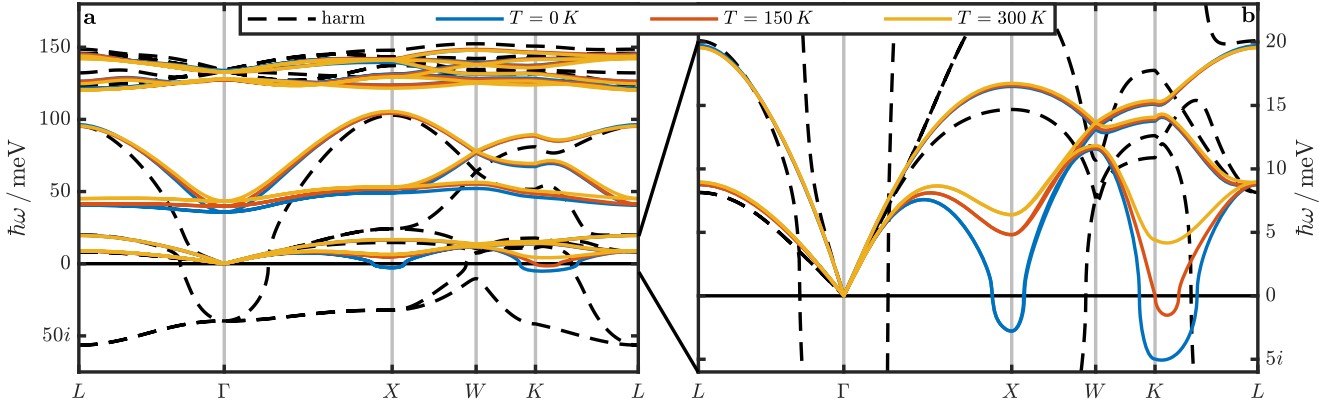

**Fig. 1 | Phonon dispersions as a function of temperature.** SSCHA phonon dispersion for different temperatures $T$ (solid colored lines). The calculations have been performed for the structure with $a = 5.040$ Å and the harmonic phonon dispersions are indicated by dashed black lines. Panel **a** shows the entire energy range and **b** only the low-energy part. The energy section for the zoomed-in region in **b** is indicated by solid black lines between the panels.

fixed by symmetry. Effectively, this means that, for a given value of the lattice constant, one can define an associated pressure within DFT ($p_{DFT}$) and within SSCHA ($p$), the latter also depending on the temperature. The fact that $p$ is, in general, larger than $p_{DFT}$, can be readily understood by considering the effects of zero-point motion[35].

To obtain a $p–T$ phase diagram, we performed several scans as a function of $p$ and $T$, described in the following: By performing a SSCHA calculation for $a = 5.040$ Å and $T = 0$ K to compare with the harmonic calculation, one finds a pressure of $p = 2.1$ GPa, i.e., very close to the synthesis pressure reported by ref. 1, and the stabilization of almost all imaginary phonon modes; except for two degenerate modes around the $X$ and the $K$ high-symmetry points, as indicated by the solid blue lines in Fig. 1. This implies that the imaginary phonon frequencies at these points in the BZ are, in fact, genuine structural instabilities.

Increasing the temperature, that instability is suppressed for $T > 150$ K (cf. Fig. 1) and $p$ increases slightly (cf. Table 1). We also observe that the optical phonon modes for energies above $\hbar\omega > 70$ meV are barely affected by temperature effects. The same applies to the acoustic modes, except for regions near the instabilities around $X$ and $K$. The optical modes around 40−50 meV harden slightly with increasing temperature. To investigate how changes in the lattice constant—or, in other words, the pressure—affect the phonon dispersions, we kept the temperature fixed at $T = 150$ K and performed SSCHA calculations for four different values of $a$. The results can be appreciated in Fig. 2, which shows the acoustic phonon branches in the low-energy region of the dispersion (for the entire dispersions as a function of pressure, we refer the reader to Supplementary Fig. 2 in Supplementary Discussion 1). As expected, the phonon energies increase with increasing pressure, and the instability is entirely absent for $p = 4.4$ GPa at 150 K.

To determine more accurately at which temperature LuH$_3$ becomes stable as a function of pressure, we have performed further SSCHA calculations, whose results we summarize in Table 1. There, DFT pressures $p_{DFT}$ and pressures $p$ at temperatures of 0, 150, and 300

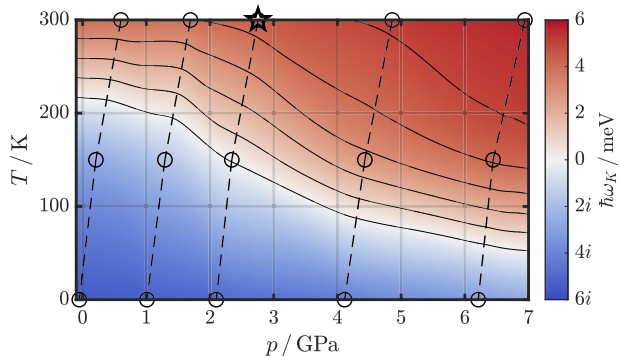

**Fig. 3 | Phase diagram of stability.** $p–T$ phase diagram of stability based on the lowest-energy phonon mode at $K$. The markers correspond to the data points reported in Table 1 and the color profile is obtained by a cubic spline interpolation. Solid lines indicate the contour lines at positive integer frequencies $\hbar\omega_K$ (in meV), and the dashed lines the isochores. The star marks the $p–T$ set for the ME calculation discussed in the main text. The corresponding dispersions are shown in Supplementary Fig. 3.

K for five different values of the lattice constant $a$ are reproduced. In addition, we present in Fig. 3 a $p–T$ phase diagram by cubic interpolation of the energy of the lowest phonon mode at $K$, indicated by circle markers, as a function of $p$ and $T$. In the following, we base our predictions about stability on the interpolated contour lines shown in Fig. 3, if not stated otherwise.

The phonon dispersions shown and discussed in the main text are based on calculations in $2 \times 2 \times 2$ supercells. The convergence with respect to the supercell size is shown and discussed in more detail in Supplementary Discussion 2, and Supplementary Figs. 6−8, and 14[51,52]. We note in passing that the calculations in higher supercells do not unambiguously answer the question of dynamic (in)stability in a specific region in the BZ in a certain pressure range. Hence, we consider the $p–T$ conditions of dynamical stability in the $2 \times 2 \times 2$ supercells as a lower limit.

Our results highlight that although quantum anharmonic lattice effects strongly influence the phonon properties, finite temperatures are imperative to stabilize the fcc LuH$_3$ phase. In fact, we find that temperatures of 80 K or higher are required to suppress the structural instability for pressures up to 6 GPa (cf. Fig. 3). As an aside, we want to note that during the revision process, we became aware of another work examining the stability of the $Fm\bar{3}m$ phase of LuH$_3$ within the SSCHA formalism[53]. We want to mention at this point that we did similar calculations for LuH$_2$, which is dynamically stable already at the harmonic level, and find that temperature and quantum anharmonic lattice effects on phonon energies are much smaller there and subordinate (more details are provided in Supplementary Fig. 13).

### Superconductivity

Having demonstrated that LuH$_3$ can become dynamically stable with the inclusion of temperature and quantum anharmonic lattice effects, we move on to obtain a detailed picture of the superconducting state, assuming the pairing is mediated by el-ph interactions. To do so, we first have to discuss its electronic properties. Unless otherwise expressed explicitly, all calculations from now on are based on the $Fm\bar{3}m$-LuH$_3$ structure with $a = 5.040$ Å (corresponding to a pressure $p$ of 2.8 GPa at room temperature). Additional cases are reported in Supplementary Table 2.

Figure 4a shows the electronic dispersions and (partial) DOS around the Fermi level, which is the energy region relevant for SC pairing within the conventional el-ph mechanism (a more zoomed-out version of this plot is provided in Supplementary Fig. 9). LuH$_3$ is a metal with three very dispersive bands crossing the Fermi level, mainly

### Table 1 | Lattice constants and pressures

| $a$ | $p_{DFT}$ | $p$(0 K) | $p$(150 K) | $p$(300 K) |
|---|---|---|---|---|
| 5.080 | −4.0 | −0.1 | 0.2 | 0.6 |
| 5.060 | −3.0 | 1.0 | 1.3 | 1.7 |
| 5.040 | −2.0 | 2.1 | 2.3 | 2.8 |
| 5.005 | 0.0 | 4.1 | 4.4 | 4.9 |
| 4.972 | 2.0 | 6.2 | 6.4 | 6.9 |

Lattice constant $a$ in Å, DFT pressure $p_{DFT}$ in GPa, and pressure $p$ in GPa within the constant-volume SSCHA calculations at different temperatures $T$.

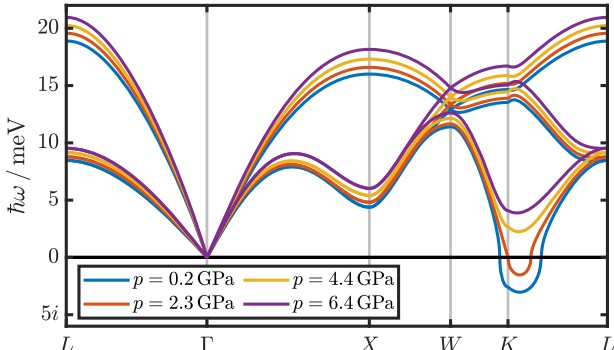

**Fig. 2 | Phonon dispersions as a function of pressure.** Low-energy part of the phonon dispersions as a function of pressure $p$ at a fixed temperature ($T = 150$ K). The full dispersion relations for all modes are provided in Supplementary Fig. 2.

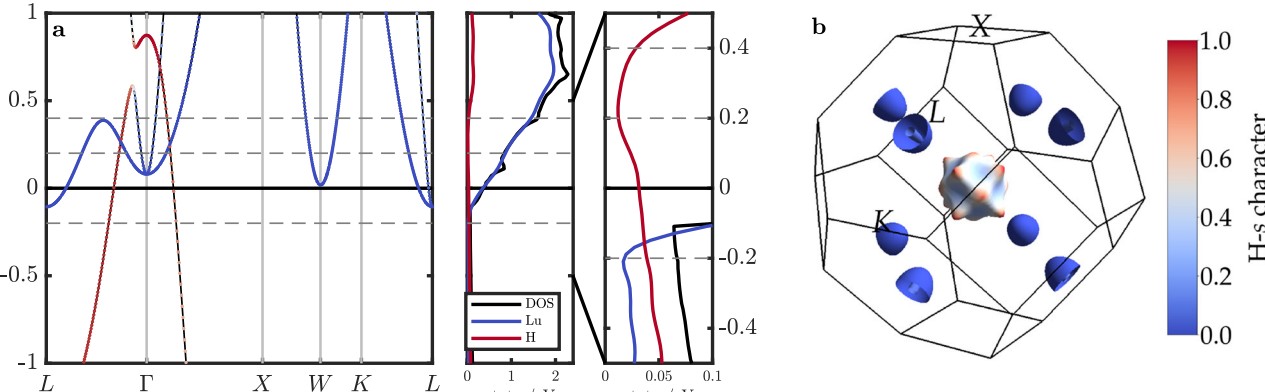

**Fig. 4 | Electronic properties. a** Electronic dispersions with H contributions indicated as colored markers, (partial) DOS for an energy window around the Fermi level, and detailed region showing the H contributions to the DOS. The dashed horizontal lines at −0.2, 0.2, and 0.4 eV indicate the energy shifts used in the rigid-band approximation calculations (more details in the main text). **b** Fermi surface with colors again representing the H contributions.

of Lu-$d$ orbital character with only little hybridization to H-$s$ states. Consequently, the Fermi surface has three small pockets: one Γ-centered hole pocket, where a small amount of H-$s$ hybridization with Lu-$d$ is observed; and two nested electron pockets around $L$ of purely Lu-$d$ character. This can also be appreciated in Fig. 4b, where we plot the Fermi surface colored according to the respective H-$s$ orbital contributions. In addition, the DOS increases steeply around the Fermi level with increasing energy, with Lu-$d$ bands above and hybridized Lu-H bands below the Fermi energy. It's worth mentioning that almost 70% of the H-$s$ states of the hole pocket around Γ originate from H located at the octahedral sites, which are unoccupied in LuH$_2$, explaining to some extent the difference in superconducting behavior.

These findings have several implications: As for example discussed in ref. 24 for the Lu-N-H system (or in general in refs. 4–9), high-$T_c$ SC requires that electrons of H-bonds participate in the el-ph pairing to harness the high phonon energies of the H vibrations. Having mainly Lu-$d$ states at the Fermi level is thus quite detrimental to RTS. Furthermore, the low density of electrons at the Fermi level is also disadvantageous for achieving the highest-$T_c$'s[4–9]. On a more technical note, the minute sizes of the Fermi surface pockets also mean that converging any calculation for el-ph interactions and SC is particularly challenging, as the BZ integrals that have to be evaluated require very dense **k** and **q** wave vector grids.

To determine the superconducting properties of LuH$_3$, we employed the Migdal-Eliashberg formalism[42] as implemented in the EPW code[43], which allows the interpolation of the electronic and vibrational properties onto very dense grids in reciprocal space using Wannier interpolation. The calculations are performed using the SSCHA-renormalized dynamical matrices at $T = 300$ K, the temperature closest to where ref. 1 report the transition, and choosing a standard value of $\mu^* = 0.1$ for the Morel-Anderson Coulomb pseudopotential[54]. (Using the corrected anharmonic dynamical matrices is justified as the ions, due to the $Fm\bar{3}m$ symmetry, are fixed to the same high-symmetry sites within DFPT and SSCHA.) With this setup, we want to determine at which temperature LuH$_3$ becomes superconducting, i.e., at which temperature the superconducting gap Δ opens.

Figure 5 summarizes the results of our calculations. Within the isotropic ME formalism, we obtain a $T_c^{\mathrm{IME}}$ of about 53 K. When solving the fully anisotropic ME equations, the superconducting critical temperature increases to about $T_c^{\mathrm{AME}} = 60$ K (see Supplementary Fig. 11). The inset of Fig. 5a shows the Fermi surface colored according to the values of the anisotropic superconducting gap function $\Delta_\mathbf{k}$ at a temperature $T = 10$ K, demonstrating a rather isotropic distribution of gap values.

To elucidate better the origin of el-ph coupling, we present in Fig. 5b the wave-vector (**q**) and mode ($v$) resolved el-ph coupling strength $\lambda_{\mathbf{q},v}$, plotted as fat bands onto the phonon dispersion. This shows clearly that almost all of the contributions to the el-ph coupling originate from the lowest two optical phonon modes and, to a lesser extent, from the lower acoustic branch, around $L$. This corresponds to the coupling of electrons between the Γ-centered hole pocket and the electron pockets at $L$, i.e., to a phonon wave vector of $\mathbf{q} = (0.5, -0.5, 0.5) \cdot 2\pi/a$ (cf. Fig. 4 and inset of Fig. 5). The respective contributions of the phonon modes to the total coupling can also be appreciated from the provided Eliashberg spectral function $\alpha^2F(\omega)$ in Fig. 5, which has most of its weight between 40–60 meV. According to our calculations, the total el-ph coupling strength $\lambda$ is about 1.7.

At this point, we want to highlight an important consequence of our results: Despite the quite sizeable $T_c$'s in the range of 50–60 K, fcc LuH$_3$ will not show SC in experiments near ambient pressures, as its critical temperature for stability is larger than 80 K in this pressure regime (cf. Fig. 3).

In the report of room-temperature SC at near-ambient pressures by ref. 1, the authors mention that LuH$_3$ could be doped with small amounts of N. To estimate the effects of changing the number of electrons on $T_c$ and to see whether room-temperature SC is attainable in the LuH$_3$ phase with such a doping, we performed additional ME calculations within the rigid-band approximation.

The energy shifts $\Delta E$ taken into consideration are indicated in Fig. 4 as dashed horizontal lines: For $\Delta E = -0.2$ eV, which corresponds to a change in electron numbers by −0.03, we find a decrease of $T_c$ to about 30 K, which is because the DOS available for pairing is relatively low in this region. This $T_c$ value refers to the isotropic calculation. In the anisotropic case, the convergence of the ME equations fails due to the low density of states at the Fermi energy, where an extremely dense **k**-grid is needed to generate explicit points at the Fermi energy. Interestingly, for positive shifts of 0.2 and 0.4 eV (corresponding to adding 0.12 and 0.33 electrons, respectively), we also observe a decrease in $T_c$ to less than 30 and 45 K, respectively. Although the DOS strongly increases for energies above the Fermi level, the H contributions, in fact, decrease (cf. Fig. 4). As these are crucial to SC in LuH$_3$, the critical temperatures consequently also decrease. More details for these calculations are provided in Supplementary Fig. 12 and Supplementary Table 2. We extend our SC analysis to pressures and phononic temperatures covered in the phase diagram in Fig. 3 and find only minor changes in $T_c$. The results are provided in Supplementary Table 2.

Despite the rigid-band approximation providing only a rough estimate for the dependence of $T_c$ with doping, it should be accurate

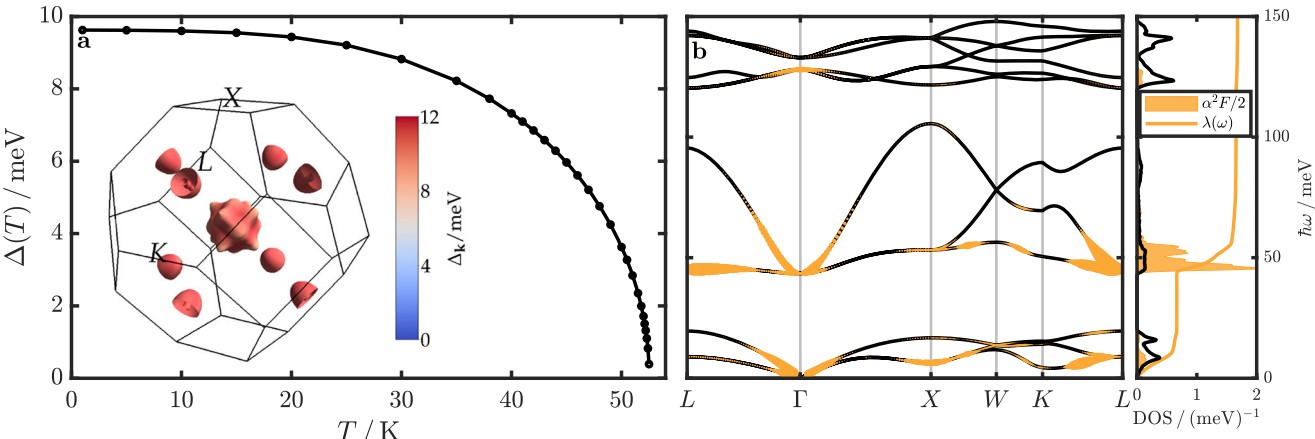

**Fig. 5 | Electron-phonon properties and superconductivity. a** Superconducting gap value Δ as a function of temperature calculated from the isotropic ME equations. The inset shows the Fermi surface colored according to the **k**-dependent gap values $\Delta_{\mathbf{k}}$ from an anisotropic ME calculation. **b** Phonon dispersion (solid black lines) as considered in the calculations with the respective wave-vector- and mode-resolved el-ph coupling strength $\lambda_{\mathbf{q},\nu}$ indicated as ochre fat bands, along with the phonon DOS (solid black line), $\alpha^2F(\omega)$, and cumulative el-ph coupling strength $\lambda(\omega)$.

enough for small doping levels. A more realistic but computationally demanding approach would be to incorporate N atoms into the LuH₃ structure, which requires dealing with large supercells. Nevertheless, the very fact that $T_c$ decreases for shifts of the Fermi level in both doping directions strongly indicates that optimizing the electron count will not be able to turn LuH₃ into a room-temperature superconductor at near-ambient pressures.

## Discussion

To summarize, our results demonstrate that even with employing quantum anharmonic corrections as implemented in SSCHA, the proposed $Fm\bar{3}m$-LuH₃ is dynamically unstable for pressures below 6 GPa at $T = 0$ K. With the additional incorporation of temperature effects, however, this phase can be stabilized for $T > 200$ K at $p = 0$ GPa ($T > 80$ K at $p = 6$ GPa). While this means that ref. 1 could indeed have synthesized the $Fm\bar{3}m$-LuH₃ phase near room-temperature, our calculations for the superconducting properties within the fully anisotropic Migdal-Eliashberg theory, including temperature- and quantum-anharmonic-corrected phonon dispersions, unambiguously demonstrate that the expected $T_c$ for an el-ph mediated SC pairing (50–60 K) is well below the reported room-temperature one; in fact, it is smaller than the critical temperature of stability in these pressure ranges, and thus $Fm\bar{3}m$-LuH₃ would decompose or transition into a different structure before it has the chance to become a (conventional) superconductor.

## Methods
### DF(P)T calculations

The electronic and harmonic phonon properties are calculated within density-functional (perturbation) theory using the plane-wave pseudopotential code Quantum Espresso[55]. We used the PBE-GGA parametrization for the exchange-correlation functional[56] in combination with scalar-relativistic optimized norm-conserving Vanderbilt pseudopotentials (ONCV)[57,58], including one valence electron for hydrogen and 25 valence electrons for Lutetium. We investigated the influence of employing different types of pseudopotentials[59] for our results, which can be found in Supplementary Discussion 2, Supplementary Table 1, and Supplementary Figs. 4 and 5. The employed Lu pseudopotential treats the 4f states explicitly and places them well below the Fermi energy in LuH₃, which is in excellent agreement with other works on explicit Lu-H-N structures[25,27,60,61]. We further confirm the 4f energy range by an all-electron calculation using WIEN2K[62]. Upon including on-site Coulomb interactions within DF(P)T + $U$ calculations, we find

no contribution of the Lu-f states on the electronic structure around the Fermi energy, and no noticeable influence on the harmonic phonon dispersions, (cf. Supplementary Figs. 15 and 16).

We performed extensive convergence tests on the numerical parameters and achieved an accuracy of below 1 meV/atom for the total energy with a 12 × 12 × 12 **k**-grid, plane-wave cutoff energy of 100 Ry, and a smearing value for the Brillouin zone integration of 0.005 Ry. The unit cell (uc) calculations were carried out in the fcc primitive unit cell with 4 atoms (lutetium in Wyckoff site 4a, hydrogen in 4b and 8c) employing a convergence threshold of $10^{-10}$ Ry for the total energy in all electronic self-consistency calculations. For the structure relaxations, we used a convergence threshold of $10^{-7}$ Ry in total energy; all force components are zero by symmetry. For the DFT calculations in 2 × 2 × 2 supercells, the **k**-grid was reduced to 6 × 6 × 6 according to the larger cell dimensions. The harmonic dynamical matrices and the self-consistent first-order variation of the potential are calculated within DFPT on a 6 × 6 × 6 **q**-grid employing a phonon self-convergency threshold of $10^{-16}$.

### SSCHA calculations

The SSCHA calculations are performed in the constant-volume mode without relaxation, as implemented in the SSCHA python package[41]. As all atomic positions are fixed by symmetry, the structure does not change in the SSCHA minimization procedure. We performed separate SSCHA calculations in 2 × 2 × 2 supercells for all 12 considered $p-T$ combinations. The physical quantities converge within a few hundred random structures, and to further reduce the stochastic noise, we increase the number of random structures in the last ensemble to 1000. The total energies, forces, and stress tensors for the individual displaced structures are obtained from DFT calculations, as described in the corresponding methods section. The SSCHA minimization is stopped for Kong-Liu ratios below 0.3 and when the ratio between the free energy gradient with respect to the auxiliary dynamical matrix and its stochastic error becomes smaller than $10^{-7}$. With these settings, the free energy differences between the last two ensembles for each $p-T$ case are well below 1 meV/uc, and the pressure differences below 0.1 GPa. The temperature-dependent anharmonic phonon dispersions are calculated from the positional free-energy Hessians in the bubble approximation, i.e., without the fourth-order corrections. As shown in Supplementary Fig. 1 in Supplementary Method 1, the fourth-order corrections are negligible in LuH₃. The largest differences in anharmonic phonon dispersions between the last two ensembles are in the order of 1 meV. The color profile and contour lines in Fig. 3 are based

on a cubic spline interpolation of the $p − T$ data points in Table 1 to a $p × T$-grid of size $150 × 150$. The final positional free energy Hessian matrices are interpolated to a $6 × 6 × 6$ **q**-grid and used as dynamical matrices for the Migdal-Eliashberg calculations. We want to stress at this point again that all data presented in the main text have been obtained using $2 × 2 × 2$ supercells and the ONCV-PBE settings described above. Further details and higher supercell calculations are provided in Supplementary Method 1 and in Supplementary Discussion 2.

## Migdal-Eliashberg calculations

We employed the EPW code package[42,43] for the Wannier interpolation of the el-ph matrix elements onto dense **k**- and **q**-grids and the subsequent self-consistent solution of the isotropic and the fully anisotropic Migdal-Eliashberg equations. In particular, we used coarse $6 × 6 × 6$ **k**- and **q** grids and interpolated onto fine $30 × 30 × 30$ reciprocal grids (see Supplementary Fig. 10), set a Matsubara frequency cutoff of 1 eV, included electronic states within $± 1$ eV around the Fermi energy, and chose a standard value for the Morel-Anderson pseudopotential of $\mu^* = 0.10$. The starting projections for the Wannierization were chosen using the SCDM method[63,64] for electronic states in the range of −15 eV to +5 eV around the Fermi energy. The rigid band shift is achieved within EPW by setting explicitly the Fermi energy to different values and hence changing the included electronic band manifold.

## Data availability

The authors confirm that the data supporting the findings of this study are available within the article and its Supplementary Information. Further information is available upon request.

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

## Acknowledgements

RL and CH acknowledge the Austrian Science Fund (FWF) and PPF acknowledges the São Paulo Research Foundation (FAPESP) under Grants 2020/08258-0 and 2021/13441-1. Calculations were performed on the Vienna Scientific Cluster and on the dcluster of the Graz University of Technology. This research was funded in part by the Austrian Science Fund (FWF) P 32144-N36. For the purpose of open access, the author has applied a CC BY public copyright licence to any Author Accepted Manuscript version arising from this submission.

## Author contributions

R.L. performed most of the calculations, P.P.F. and M.A. assisted with additional calculations and figure design, and C.H. supervised the project. All authors contributed to the discussion of the results and participated in preparing the manuscript.

## Competing interests

The authors declare no competing interests.
