## [Peer Review File · Nature Communications]

Temperature and quantum anharmonic lattice effects on stability and superconductivity in lutetium trihydrideREVIEWER COMMENTS:

Reviewer 1:

In this manuscript, Roman Lucrezi *et al.* considered temperature and quantum anharmonic lattice effects on the structural stability and superconductivity of lutetium trihydrides. The results suggested that $Fm\bar{3}m$ LuH₃ is dynamically unstable for pressures below 6 GPa at T = 0 K, but it can be stabilized at P = 6 GPa when T > 80 K (P = 0 GPa, T > 200 K) with the additional incorporation of temperature effects. The calculated T_c for $Fm\bar{3}m$ LuH₃ at 2.8 GPa within the Migdal-Eliashberg formalism is 50-60 K, which is much lower than the reported room-temperature superconductivity and even lower than the temperature required for a stable lattice.

The results are original, attractive and instructive. However, the low superconductivity in LuH₃ does not mean that N-doped LuH₃ has the similar behavior. If the authors can do the same work for N-doped LuH₃, which will be helpful for readers to understand the truth of the superconductivity in Lu-N-H.

1. Is the P-T phase diagram in Figure 3 obtained by fitting the data in Table I? In the fourth paragraph of page 2, the authors mention that “we show that the $Fm\bar{3}m$ LuH₃ phase can be stabilized at temperatures above 170 K for pressures of 1 GPa”. If the P-T phase diagram is fitted, I think it should be explained in the manuscript and the authors should use SSCHA software to calculate the phonon dispersion at 170 K for 1 GPa to verify the reliability of this conclusion.

2. In Figure 4, the contributions of H are multiplied with a factor of 5 to improve visibility. But it is easy to leave the readers with the vague impression that H contributes a lot. It is suggested to appropriately modify the coordinate range instead of multiplying the coefficient of 5 to indicate the contributions of H.

3. The correctness of the statement in the fifth paragraph of the Introduction that the hexagonal $P6_3$ structure is the most thermodynamically stable one for the LuH₃ stoichiometry has yet to be checked by the authors.

4. The authors should give a graph like the one in Figure 1 of the ref.5 cited in the Supplemental Material. In the figure, the reliability of the machine learning potential used for calculations can be verified by comparing the total energy or phonon spectra.

5. In Table 1, all the results are calculated with $2 \times 2 \times 2$ supercells. In Fig. S7, why did the authors use a different $4 \times 4 \times 4$ supercell to calculate phonon dispersions for temperatures ≥ 300 K and pressures above 2.8 GPa and 6.9 GPa?

Reviewer #2 (Remarks to the Author):

In this work, the authors provide unique insights into the superconducting origin of Lu-N-H system, reported by Dasenbrock-Gammon et al [Nature 615, 244 (2023)]. In detail, the authors have constructed the P-T phase diagram with the inclusion of temperature, pressure, and quantum anharmonic lattice effects. Then, they calculated the superconducting properties within the fully anisotropic Migdal-Eliashberg theory including temperature- and quantum-anharmonic-corrected phonon dispersions. Their results unambiguously demonstrate that pure or doped Fm3m-LuH3 is not the candidate structure proposed in the experiment. Their results not only resolve conflicting experimental and theoretical findings of Fm3m-LuH3 but also are of great significance to accurately describe the stability and superconductivity of hydrides. I recommend publishing this work.

Reviewer #3 (Remarks to the Author):

Perhaps the two biggest results in high pressure physics & superconductivity have come from Ranga Dias' group & collaborators: the controversial and ultimately retracted work on C-S-H, and a very recent claim of near-ambient room temperature superconductivity in nitrogen-"doped" lutetium hydride. These latter claims are so striking that the high pressure community is hard at work to replicate (experimentally) or understand (theoretically).

One key difficulty in lutetium hydride superconductivity is that the optical properties reported by several groups support LuH2. But calculations show LuH2 cannot be a BCS superconductor. On the other hand Fm-3m LuH3, which would match experimental diffraction patterns too, is highly unstable at the harmonic level. This is actually promising, because hydrides such as LaH10 are very unstable at the harmonic level, but (quantum) anharmonic corrections stabilise it and help superconduct near room temperature.

In this work, Lucrezi and collaborators present a computational/theoretical analysis on LuH3, assessing whether it is the superconducting material claimed by Dasenbrock-Gammon et al. in Ref. 1. I'll start by saying that, due to the extremely high interest in the system at hand, the research topic is very timely. I am right now in AIRAPT/EHPRG in Edinburgh, where several talks have been devoted to this very system. Sadly, the authors of the current work are not here to discuss their exciting work in person.

Here, Lucrezi et al. show that LuH3 cannot be a room temperature BCS superconductor. In order to do so, they have performed state of the art lattice dynamics simulations, beyond the harmonic level, by using the SSCHA technique pioneered by Errea in order to analyse anharmonic effects. These have been shown to be crucial in several hydrides, both standard metals and superconductors. One of their results is that indeed anharmonic effects, coupled with temperature and pressure tuning, do indeed stabilise LuH3, which was shown before to be highly unstable in the harmonic regime. However, they also show that LuH3 metallic character comes from Lu-bands coupled mostly with Lu and Lu-H modes. The authors also show calculations of the superconducting critical temperature. This is ~60 K, normally very high but nowhere near enough room temperature. The authors also find that, assuming doping barely changes the band structure, small nitrogen impurities should not affect Tc very significantly. The main message is: LuH3 + N_x as a BCS superconductor cannot explain observations.

Overall, I'm enthusiastic about the quality and timeliness of the work shown. I also believe the paper provides most of the data I'd need to reproduce it. However, I do have a few questions that I'd like the authors to address.

0. While the authors have done a variety of calculations. I assume all graphs shown in the main paper are the PBE DFT, with the ONC pseudopotential, on the 2x2x2 SSCHA grid. Could the authors confirm this? Less importantly, in the SM, does PZ ONCP mean "the Perdew-Zunger

pseudopotential, but PBE runs" or "PZ pseudopotential _and_ exchange correlation functional for the whole calculations"?

1. Regarding the phonon stability, dispersions shown in the main article omit the Gamma-K line. However, in the supplementary material, Fig. S4 shows a clear instability halfway through. This is irrelevant of the approximation. Is this instability real, is it a cause of the SSCHA grid being $2 \times 2 \times 2$ and thus not being able to consider the q-point halfway through, or something else? The MLPP phonon dispersions later on do show difficulty converging the phonon spectrum.

2. Their calculations seem to suggest that the choice of pseudopotential may be as important as the functional. This would seem to imply that at least one set of pp's isn't correct. The authors choose ONCP based on lattice parameter. Is there any other qualitative difference? Also, what PAW pseudopotential was used? Wentzcovitch group's? Or an inhouse one? In that case, what were roughly the parameters of choice? In the same vein, are the different phonon results between PP's in the section above just a case of the DFT pressure being different? Would I get similar results at similar computed pressures?

3. The Migdal-Eliashberg T_c calculations have been performed with the phonon spectrum at 2 GPa & 300 K, but these produce a much lower T_c . This is, at the very least, inconsistent. One data point cannot also address the strongly parabolic trend reported for T_c experimentally. Would it be possible to obtain the analysis at a different pressure and temperature? (e.g. 4 GPa & 150 K). I understand EPW calculations aren't the cheapest.

4. I do not understand Fig. S6. I can clearly appreciate the difficulty converging the Gamma-K branch. However, the difference in X, especially at 2.8 GPa, is very baffling. This is a point that should be included in all the even-q grids. However, the denser the grid, the more unstable it becomes. Why would that be? Is this pointing towards non-smooth behaviour in the MTP potentials? Or is this an extrapolation issue?

5. Perhaps the elephant in the room, do the authors expect correlation effects in Lu 4f electrons to be important? If one used self-consistent +U to compute, e.g. harmonic phonons at the same lattice parameters, would one get very different results?

Responses to Reviewers

Reviewer 1

In this manuscript, Roman Lucrezi et al. considered temperature and quantum anharmonic lattice effects on the structural stability and superconductivity of lutetium trihydrides. The results suggested that $Fm\bar{3}m$ LuH_3 is dynamically unstable for pressures below 6 GPa at $T = 0$ K, but it can be stabilized at $P = 6$ GPa when $T > 80$ K ($P = 0$ GPa, $T > 200$ K) with the additional incorporation of temperature effects. The calculated T_c for $Fm\bar{3}m$ LuH_3 at 2.8 GPa within the Migdal-Eliashberg formalism is 50-60 K, which is much lower than the reported room-temperature superconductivity and even lower than the temperature required for a stable lattice. The results are original, attractive and instructive.

We'd like to thank the Referee for the assessment and will answer all open questions point-by-point in the following.

However, the low superconductivity in LuH_3 does not mean that N-doped LuH_3 has the similar behavior. If the authors can do the same work for N-doped LuH_3 , which will be helpful for readers to understand the truth of the superconductivity in Lu-N-H.

Examining slight N-doping explicitly is extremely challenging, as large supercells would be needed to simulate the doping in a proper, quasi-random fashion. Due to the computational expense of both SSCHA and EPW, the treatment of such supercells is currently not feasible. In order to at least get an approximate idea on the robustness/sensitivity of T_c upon electron and hole doping, we explored the effects of rigidly shifting the Fermi level on the critical temperature.

Our results show that T_c actually decreases upon both positive and negative Fermi level shifts. In the course of this reply, we have extended our superconductivity (SC) analysis to other pressures and temperatures (see answer to Reviewer 3/3) and find that T_c never exceeds ~ 60 K within this approximation.

Other works that have incorporated effects of N-doping via the virtual crystal approximation [1] or supercell approaches [2, 3], for example, arrive at similar conclusions, i.e., while a slight increase in T_c might be possible upon doping, the $Fm\bar{3}m$ $\text{LuH}_3/\text{LuH}_2$ (parent) system cannot support room-temperature superconductivity at ambient pressure within an electron-phonon coupling picture.

1.) Is the P - T phase diagram in Figure 3 obtained by fitting the data in Table I? In the fourth paragraph of page 2, the authors mention that "we show that the $Fm\bar{3}m$ LuH_3 phase can be stabilized at temperatures above 170 K for pressures of 1 GPa". If the P - T phase diagram is fitted, I think it should be explained in the manuscript and the authors should use SSCHA software to calculate the phonon dispersion at 170 K for 1 GPa to verify the reliability of this conclusion.

The phase diagram presented in Figure 3 is indeed based on a cubic spline interpolation of the explicit data points shown as circle markers and stated in Table I, and the values mentioned by the Referee are based on this interpolation. We thank the Referee for pointing out that this has not been mentioned explicitly in the main text and have added a note to the corresponding paragraphs in the text.

In order to increase the interpolation reliability in the mentioned $p - T$ region, we performed three additional SSCHA calculations for a lattice constant of $a = 5.06$ Å and 0, 150, and 300 K, corresponding to pressures of 1.0, 1.3, and 1.7 GPa, respectively. The refined interpolation of the phase diagram corrects our previous estimate of 170 K to 200 K.

We added a corresponding sentence to the section "Methods/SSCHA calculations" and to the caption of Figure 3. We have updated Table I, Figure 3, and Figure S3 (Supplemental Material) accordingly and provide them here for your convenience:

Table I: Lattice constant a in Å, DFT pressure p_{DFT} in GPa, and pressure p in GPa within the constant-volume SSCHA calculations at different temperatures.

a	p_{DFT}	$p(0\text{ K})$	$p(150\text{ K})$	$p(300\text{ K})$
5.080	-4.0	-0.1	0.2	0.6
5.060	-3.0	1.0	1.3	1.7
5.040	-2.0	2.1	2.3	2.8
5.005	0.0	4.1	4.4	4.9
4.972	2.0	6.2	6.4	6.9

Figure 3: p - T phase diagram of stability based on the lowest-energy phonon mode at K . The markers correspond to the data points reported in Tab. I and the colour profile is obtained by a cubic spline interpolation. Solid lines indicate the contour lines at positive integer frequencies $\hbar\omega_K$ (in meV) and the dashed lines isochores. The star marks the p - T set for the ME calculation discussed in the main text. The corresponding dispersions are shown in Fig. S3.

Figure S3: Low-energy range of the phonon dispersion for all considered p - T combinations in the $2 \times 2 \times 2$ supercell. The harmonic phonon dispersion for the corresponding lattice constant a at $T = 0$ K are shown as dashed lines.

2.) In Figure 4, the contributions of H are multiplied with a factor of 5 to improve visibility. But it is easy to leave the readers with the vague impression that H contributes a lot. It is suggested to appropriately modify the coordinate range instead of multiplying the coefficient of 5 to indicate the contributions of H.

We thank the Referee for this suggestion and have modified Fig. 4(a) accordingly, as reproduced below.

Figure 4(a): Electronic dispersions with H contributions indicated as colored markers, (partial) DOS for an energy window around the Fermi level, and detailed region showing the H contributions to the DOS. The dashed horizontal lines at -0.2, 0.2, and 0.4 eV indicate the energy shifts used in the rigid-band approximation calculations (more details in the main text).

The correctness of the statement in the fifth paragraph of the Introduction that the hexagonal $P\bar{6}_3$ structure is the most thermodynamically stable one for the LuH_3 stoichiometry has yet to be checked by the authors.

We thank the Referee for raising this point. We were basing our statement on another recent work of some of the current authors, where crystal structure prediction searches for the full ternary phase space were performed (Ferreira *et al.* [4]), identifying hexagonal $P\bar{6}_3$ - LuH_3 to be the most thermodynamically stable structure for the LuH_3 stoichiometry at 0 GPa and 0 K. Other recent works on the thermodynamic stability in the Lu–H–N system, however, report $P\bar{3}c1$ - LuH_3 as the thermodynamically most stable LuH_3 prototype (Hilleke *et al.*[2], Xie *et al.* [5] and Fang *et al.*[3]). Conversely, Liu *et al.* [6], find $R\bar{3}2$ - LuH_3 to be on the convex hull. We attribute this discrepancy among the results to the very small energy differences between the respective structures, meaning that different codes and pseudopotentials employed can lead to a slightly different ranking between those low-energy structures. All works, however, agree upon the fact that the $Fm\bar{3}m$ structure is not the thermodynamically most stable phase for LuH_3 at 0 GPa and 0 K.

To avoid any ambiguity or misinterpretation, we have revised the corresponding sentence accordingly: ‘While these theoretical works have confirmed the presence of the $Fm\bar{3}m$ - LuH_2 phase on the convex hull, they have also revealed that for LuH_3 the cubic $Fm\bar{3}m$ phase is not the thermodynamically most stable one close to ambient pressure.’

The authors should give a graph like the one in Figure 1 of the Ref. 5 cited in the Supplemental Material. In the figure, the reliability of the machine learning potential used for calculations can be verified by comparing the total energy or phonon spectra.

We thank the Referee for this suggestion. While we do state the quality of the obtained MTPs in the SM, we agree that an additional graphical representation can only be beneficial to the clarity and transparency of this work. We have therefore included the Figure S14 below in the SM.

Figure S14: MTP validation for the predicted energies E_{MTP} , force components F_{MTP} , and stress tensor components σ_{MTP} of the individuals generated in the DFT SSCHA calculations for lattice constants of (a) 5.040 Å, (b)-(c) 4.972 Å, and (d) 4.915 Å. The MTPs corresponding to (a)-(d) are used to obtain the SSCHA results shown in Fig. S6 and S7, the MTPs corresponding to (b) and (c) are used to obtain the SSCHA results shown in Fig. S8. Each subpanel shows the MTP prediction versus the corresponding DFT values. The energies are plotted with respect to the DFT total energy of the undisplaced structure E_{ref} with the indicated lattice constant. The diagonal elements of the stress tensor σ_{ii} are plotted as blue dots, the off-diagonal elements $\sigma_{i \neq j}$ as green dots. The diagonal black line represents the reference for a perfect prediction.

In Table 1, all the results are calculated with $2 \times 2 \times 2$ supercells. In Fig. S7, why did the authors use a different $4 \times 4 \times 4$ supercell to calculate phonon dispersions for temperatures ≥ 300 K and pressures above 2.8 GPa and 6.9 GPa?

The Referee is right in stating that our main results are based on DFT calculations in $2 \times 2 \times 2$ supercells. As SSCHA is a supercell-based method, the convergence of any result obtained with it has to be checked with respect to the supercell size. We present the outcome of our supercell convergence tests in Figures S6-S8 of the SM for full transparency.

These checks were done by employing machine-learned moment tensor potentials (MTP) to be able to consider larger cells. We do not find any particularly striking supercell effect on the overall shape of the phonon dispersion, besides some fragile low-energy features that we discuss in detail in the Supplemental Material. In particular, in Fig. S7 we examine the behaviour of the inconclusive convergence of the low-energy modes at X and on $\Gamma - K$ for 2.8 and 6.9 GPa at higher temperatures than considered in the main text and in Fig. S6. In order to save on computational resources, we did this for a $4 \times 4 \times 4$ and not the $6 \times 6 \times 6$ supercell. For completeness, we added the same analysis in a $2 \times 2 \times 2$ supercell as subpanels (a) and (b) to Figure S7:

Figure S7: Low-energy phonon dispersions obtained in (a)-(b) $2 \times 2 \times 2$ and (c)-(d) $4 \times 4 \times 4$ supercells at $T \geq 300$ K and pressures above (a),(c) 2.8 GPa and (b),(d) 6.9 GPa. The indicated pressures refer to the value obtained at $T = 300$ K. For a fixed lattice constant, higher temperatures lead to a higher pressure. The SSCHA individuals are evaluated with different MTPs trained on ONCV-PBE calculations with unit-cell lattice constants (a),(c) 5.040 Å, and (b),(d) 4.972 Å.

Reviewer 2

In this work, the authors provide unique insights into the superconducting origin of Lu–N–H system, reported by Dasenbrock-Gammon et al [Nature 615, 244 (2023)]. In detail, the authors have constructed the P – T phase diagram with the inclusion of temperature, pressure, and quantum anharmonic lattice effects. Then, they calculated the superconducting properties within the fully anisotropic Migdal-Eliashberg theory including temperature- and quantum-anharmonic-corrected phonon dispersions. Their results unambiguously demonstrate that pure or doped $Fm\bar{3}m$ - LuH_3 is not the candidate structure proposed in the experiment. Their results not only resolve conflicting experimental and theoretical findings of $Fm\bar{3}m$ - LuH_3 but also are of great significance to accurately describe the stability and superconductivity of hydrides.

I recommend publishing this work.

*Guochun Yang
Yanshan University*

We thank Prof. Guochun Yang for his efforts, for expressing appreciation for our work, and for the recommendation to publish our article in Nature Communications.

Reviewer 3

Perhaps the two biggest results in high pressure physics & superconductivity have come from Ranga Dias' group & collaborators: the controversial and ultimately retracted work on C–S–H, and a very recent claim of near-ambient room temperature superconductivity in nitrogen-"doped" lutetium hydride. These latter claims are so striking that the high pressure community is hard at work to replicate (experimentally) or understand (theoretically).

One key difficulty in lutetium hydride superconductivity is that the optical properties reported by several groups support LuH_2 . But calculations show LuH_2 cannot be a BCS superconductor. On the other hand $Fm\bar{3}m$ LuH_3 , which would match experimental diffraction patterns too, is highly unstable at the harmonic level. This is actually promising, because hydrides such as LaH_{10} are very unstable at the harmonic level, but (quantum) anharmonic corrections stabilise it and help superconduct near room temperature.

In this work, Lucrezi and collaborators present a computational/theoretical analysis on LuH_3 , assessing whether it is the superconducting material claimed by Dasenbrock-Gammon et al. in Ref. 1. I'll start by saying that, due to the extremely high interest in the system at hand, the research topic is very timely. I am right now in AIRAPT/EHPRG in Edinburgh, where several talks have been devoted to this very system. Sadly, the authors of the current work are not here to discuss their exciting work in person.

Here, Lucrezi et al. show that LuH_3 cannot be a room temperature BCS superconductor. In order to do so, they have performed state of the art lattice dynamics simulations, beyond the harmonic level, by using the SSCHA technique pioneered by Errea in order to analyse anharmonic effects. These have been shown to be crucial in several hydrides, both standard metals and superconductors. One of their results is that indeed anharmonic effects, coupled with temperature and pressure tuning, do indeed stabilise LuH_3 , which was shown before to be highly unstable in the harmonic regime. However, they also show that LuH_3 metallic character comes from Lu-bands coupled mostly with Lu and Lu–H modes. The authors also show calculations of the superconducting critical temperature. This is ~ 60 K, normally very high but nowhere near enough room temperature. The authors also find that, assuming doping barely changes the band structure, small nitrogen impurities should not affect T_c very significantly. The main message is: $\text{LuH}_3 + \text{N}_x$ as a BCS superconductor cannot explain observations.

Overall, I'm enthusiastic about the quality and timeliness of the work shown. I also believe the paper provides most of the data I'd need to reproduce it. However, I do have a few questions that I'd like the authors to address.

We would like to thank the Referee for the assessment and want to respond to all comments point-by-point in the following.

0. While the authors have done a variety of calculations. I assume all graphs shown in the main paper are the PBE DFT, with the ONC pseudopotential, on the $2 \times 2 \times 2$ SSCHA grid. Could the authors confirm this? Less importantly, in the SM, does PZ ONCP mean "the Perdew-Zunger pseudopotential, but PBE runs" or "PZ pseudopotential and exchange correlation functional for the whole calculations"?

The Referee is right in saying that all data presented in the main text are obtained with ONCV PBE pseudopotentials for DFT calculations and on $2 \times 2 \times 2$ supercells for SSCHA. In order to avoid any ambiguity, we added the following paragraph to the method section of the main text: "We want to stress at this point again that all data presented in the main text have been obtained using $2 \times 2 \times 2$ supercells and the ONCV-PBE settings described above."

As for the second part of the Referee's comment: The labels "-PZ" and "-PBE" refer to both, pseudopotential generation and exchange-correlation functional form in the actual DFT calculation. So, for example, "ONCV-PZ" refers to a PZ-LDA functional of ONCV type for the whole calculation. The ONCV-PZ pseudopotentials are generated using the ONCVSP code and match the settings of the SG15 collection, described in [7, 8]. We added the following sentence to the related paragraph in the Supplemental Material: "In the following, the labels '-PZ' and '-PBE' refer to both the pseudopotential generation and the exchange-correlation functional form in the actual DFT calculation. The ONCV-PZ pseudopotentials are generated using the ONCVSP code and match the settings of the SG15 collection, described in [7, 8]."

1. Regarding the phonon stability, dispersions shown in the main article omit the $\Gamma-K$ line. However, in the supplementary material, Fig. S4 shows a clear instability halfway through. This is irrelevant of the approximation. Is this instability real, is it a cause of the SSCHA grid being $2 \times 2 \times 2$ and thus not being able to consider the q -point halfway through, or something else? The MLPP phonon dispersions later on do show difficulty converging the phonon spectrum.

From Fig. S4, i.e. in the $2 \times 2 \times 2$ supercell case, it is indeed inconclusive, as no point along $\Gamma - K$ or even in its vicinity is captured explicitly, and the apparent instability could also prove to be just an interpolation artifact. To shed light on this, we present in the SM the results of SSCHA calculations in larger supercells (Supplemental Figures S5 and S6). As detailed below in our response to the Referee's questions about Supplemental Figure S6, we find an intricate behaviour with respect to the supercell size that doesn't seem to be fully converged even for large $6 \times 6 \times 6$ cells.

2. Their calculations seem to suggest that the choice of pseudopotential may be as important as the functional. This would seem to imply that at least one set of pp's isn't correct. The authors choose ONCP based on lattice parameter. Is there any other qualitative difference? Also, what PAW pseudopotential was used? Wentzcowitch group's? Or an inhouse one? In that case, what were roughly the parameters of choice? In the same vein, are the different phonon results between PP's in the section above just a case of the DFT pressure being different? Would I get similar results at similar computed pressures?

The PAW potentials employed were taken from the *pslibrary* by Andrea Dal Corso [9], we added the reference to the main manuscript.

We would argue that, overall, the PBE results agree very well with each other, except for strongly anharmonic modes. In order to convey this better, we have modified Supplemental Fig. S4(a), where we compare the SSCHA phonon dispersions for ONCV-PBE (blue) and PAW-PBE (red) for the same lattice constant. These curves are almost on top of each, except around Γ , X , and L , where strong anharmonic behaviour means that the results are extremely sensitive to the computational details. One difference between ONCV-PBE and PAW-PBE is, however, that PAW-PBE gives slightly lower values for the pressure than ONCV-PBE.

In the same figure, we also show the SSCHA phonon dispersion for ONCV-PZ for the same lattice constant, which, apart from the low-frequency modes below 20 meV, are systematically shifted to lower frequencies (by about 5-10 meV) compared to PBE. ONCV-PZ also gives a considerably lower pressure than the PBE pseudopotentials (about 5-7 GPa less).

To answer the last question of the Referee, we have included the new Supplemental Figure S4(b), where we compare SSCHA phonon dispersions for ONCV-PBE and ONCV-PZ pseudopotentials relaxed to a similar SSCHA pressure. In this case the ONCV-PZ results are systematically shifted to higher frequencies compared to the ONCV-PBE calculations. We have revised Figure S4(a) and added Figure S4(b) to the SM in combination with the following paragraph: "We point out that considering the substantially lower pressure value obtained with ONCV-PZ at fixed a , the phonon frequencies match the PBE results well. In Figure S4(b), we show the effect of balancing the pressure values in an ONCV-PZ calculation with a reduced lattice constant of $a = 4.914 \text{ \AA}$."

Figure S4: (a) SSCHA phonon dispersion obtained with different functional and PP form in the $2 \times 2 \times 2$ supercell at $T = 300$ K and $a = 5.040$ Å. The pressures obtained in SSCHA are indicated in the legend. The corresponding DFT pressures are -2.0, -3.9, and -9.5 GPa for ONCV-PBE, PAW-PBE, and ONCV-PZ, respectively. (b) Comparison of ONCV-PBE (blue) and ONCV-PZ (green) with a similar calculated pressure.

3. The Migdal-Eliashberg T_c calculations have been performed with the phonon spectrum at 2 GPa & 300 K, but these produce a much lower T_c . This is, at the very least, inconsistent. One data point cannot also address the strongly parabolic trend reported for T_c experimentally. Would it be possible to obtain the analysis at a different pressure and temperature? (e.g. 4 GPa & 150 K). I understand EPW calculations aren't the cheapest.

We thank the Referee for the suggestion to examine the superconducting behaviour for more points in the $p - T$ phase diagram than just for a single point. We extended our analysis to three different lattice constants and two different phononic temperatures, and gladly present our results in Table S2. The table is added to the Supplemental Material and a corresponding sentence is added to the main text.

Table S2: Overview of the performed EPW calculations and obtained critical temperatures T_c within isotropic (anisotropic) Migdal-Eliashberg equations, denoted by IME (AME). The calculations are performed for three different lattice constants and two SSCHA temperatures, corresponding to the pressures given in Tab. I in the main text. The Fermi energy shift ΔE used in the rigid-band approximation is given in eV, the critical temperature T_c in K. The dynamical matrices are obtained from the SSCHA calculations on a $2 \times 2 \times 2$ supercell interpolated onto a $6 \times 6 \times 6$ \mathbf{q} -grid, as described in the main text. For the cases denoted by a dagger (\dagger), the interpolation to a dense \mathbf{q} -grid leads to imaginary phonon frequencies along $\Gamma - K$, which are set to zero in the EPW calculations.

ΔE	phonons at 150 K						phonons at 300 K					
	2.3 GPa \dagger		4.4 GPa \dagger		6.4 GPa		2.8 GPa		4.9 GPa		6.9 GPa	
	T_c^{IME}	T_c^{AME}	T_c^{IME}	T_c^{AME}	T_c^{IME}	T_c^{AME}	T_c^{IME}	T_c^{AME}	T_c^{IME}	T_c^{AME}	T_c^{IME}	T_c^{AME}
-0.2	36	10	37	12	33	14	31	6	30	9	29	10
0.0	56	64	56	61	52	54	53	60	51	57	43	50
0.2	24	36	20	29	16	25	23	33	18	27	15	23
0.4	39	49	30	38	24	33	38	48	27	37	23	32

4. I do not understand Fig. S6. I can clearly appreciate the difficulty converging the $\Gamma-K$ branch. However, the difference in X , especially at 2.8 GPa, is very baffling. This is a point that should be included in all the even- q grids. However, the denser the grid, the more unstable it becomes. Why would that be? Is this pointing towards non-smooth behaviour in the MTP potentials? Or is this an extrapolation issue?

It is true that X is commensurate with all even \mathbf{q} -grids or -supercells (as well as L and Γ). However, in SSCHA, the modes at different \mathbf{q} -points are not independent of each other, like they are in the usual harmonic approximation, but rather incorporate coupling effects to all other modes. In that sense, SSCHA results for modes at \mathbf{q} -points commensurate with different supercell sizes are not expected to be the same for all supercell sizes, i.e. the supercell size is a parameter that needs to be converged. The behaviour of the specific mode at X suggests that it couples strongly to many other modes. This is definitely a very interesting topic, but within our current computational resources, we could not converge that mode further than to $6 \times 6 \times 6$ supercells, and hence we can not really make any conclusion on the converged value (which might also rise again). The same argument holds for the lowest mode on the path $\Gamma - K$. We trained our MTPs on equal amounts of structures in $2 \times 2 \times 2$, $3 \times 3 \times 3$, and $4 \times 4 \times 4$ supercells, so we do not expect large errors for higher supercells. And in fact, we did not notice any heavy outliers or unexpected trends in the MTP predictions for high supercells.

In light of this and the arguments mentioned in the responses to the Referee's questions 1 and 2, we want to state that the calculations in higher supercells do not unambiguously answer the question of dynamic (in)stability in a specific region in the BZ in a certain pressure range, which is why we consider the $p-T$ conditions of dynamical stability in the $2 \times 2 \times 2$ supercells as a *best case* scenario, i.e. a lower limit. A corresponding statement is given as note in Ref. [53] of the main text.

5. Perhaps the elephant in the room, do the authors expect correlation effects in Lu 4f electrons to be important? If one used self-consistent $+U$ to compute, e.g. harmonic phonons at the same lattice parameters, would one get very different results?

This is a very interesting point that we gladly elaborate more on. Lu has completely filled 4f states that in our calculations (employing ONCV-PBE pseudopotentials) give rise to highly localized, low-dispersive bands below -4 eV with virtually zero contribution in the close vicinity of the Fermi level. In order to rule out any bias from the pseudopotential (PP), we have confirmed this result by additional all-electron (AE) WIEN2K calculations, which place the 4f states at very similar energies, as shown in Figure S15(c). This result is in excellent agreement with other works on explicit Lu-H-N structures, which all have the 4f states in a sizeable energy range below the Fermi energy [2, 3, 10, 11].

As there is no hybridization between the 4f manifold and the Fermi surface H-s or Lu-d states, the on-site Coulomb interactions of the f electrons do not play a crucial role for SC or stability. To demonstrate this, we have performed additional DFT+ U calculations with $U = 4, 6, \text{ and } 8 \text{ eV}$. As can be appreciated in Figure S15, the additional interaction pushes the f-states manifold further down and away from the Fermi level. In Figure S16, we present a comparison between harmonic phonon dispersions with and without U , demonstrating that the changes in the phonon dispersion upon adding a Hubbard interaction via DFPT+ U are negligible.

We have added Figures S15 and S16 to the SM and added the following paragraph to the Methods sections: “The employed Lu pseudopotential treats the 4f states explicitly and places them well below the Fermi energy in LuH₃, which is in excellent agreement with other works on explicit Lu-H-N structures [2, 3, 10, 11]. We further confirm the 4f energy range by an all-electron calculation using WIEN2K [12]. Upon including on-site Coulomb interactions within DF(P)T+U calculations, we find no contribution of the Lu-f states on the electronic structure around the Fermi energy, and no noticeable influence on the harmonic phonon dispersions, (cf. Fig. S15 and S16 in the SM).”

Figure S15: (a) Electronic dispersion and (b) corresponding DOS for LuH₃ with a lattice constant of $a = 5.040 \text{ \AA}$ for a regular DFT calculation within the ONCV-PBE setting (black line) and for DFT+U calculations with $U = 4, 6, \text{ and } 8 \text{ eV}$ for the Lu- f states (coloured lines). The different lines lie on top of each other for energy values around and above the Fermi energy E_F . (c) DOS (lines) and partial (p) DOS (shaded areas) for the Lu-4f states with $U = 0 \text{ eV}$ calculated with pseudopotentials (PP) within the ONCV-PBE setting and in an all-electron (AE) calculation with WIEN2K. For the latter, the size of the LAPW basis set was determined by $R \cdot K_{\text{max}} = 7.0$ and muffin-tin radii were chosen as $2.50 a_0$ and $1.43 a_0$ for Lu and H, respectively. We used PBE as xc-potential and 1059 irreducible \mathbf{k} -points were used for the BZ sampling.

Figure S16: Harmonic phonon dispersions on a $2 \times 2 \times 2$ q -grid for LuH_3 with a lattice constant of $a = 5.040 \text{ \AA}$ for a regular DFPT calculation within the ONCV-PBE setting (dashed black line) and for a DFPT+ U calculation with $U = 4 \text{ eV}$ for the Lu- f states (solid blue line). The maximum differences in the two dispersion are $< 0.5 \text{ meV}$.

References

- [1] Z. Huo, D. Duan, T. Ma, Z. Zhang, Q. Jiang, D. An, H. Song, F. Tian, and T. Cui, "First-principles study on the conventional superconductivity of N-doped fcc- LuH_3 ," *Matter and Radiation at Extremes*, vol. 8, p. 038402, 05 2023.
- [2] K. P. Hilleke, X. Wang, D. Luo, N. Geng, B. Wang, F. Belli, and E. Zurek, "Structure, stability, and superconductivity of N-doped lutetium hydrides at kbar pressures," *Phys. Rev. B*, vol. 108, p. 014511, Jul 2023.
- [3] Y.-W. Fang, Đ. Dangić, and I. Errea, "Assessing the feasibility of near-ambient conditions superconductivity in the Lu-N-H system," *arXiv preprint arXiv:2307.10699*, 2023.
- [4] P. P. Ferreira, L. J. Conway, A. Cucciari, S. D. Cataldo, F. Giannessi, E. Kogler, L. T. F. Eleno, C. J. Pickard, C. Heil, and L. Boeri, "Search for ambient superconductivity in the Lu-N-H system," *Nature Communications*, vol. 14, no. 5367, 2023.
- [5] F. Xie, T. Lu, Z. Yu, Y. Wang, Z. Wang, S. Meng, and M. Liu, "Lu-H-N phase diagram from first-principles calculations," *Chinese Physics Letters*, vol. 40, p. 057401, apr 2023.
- [6] M. Liu, X. Liu, J. Li, J. Liu, Y. Sun, X.-Q. Chen, and P. Liu, "Parent structures of near-ambient nitrogen-doped lutetium hydride superconductor," *Phys. Rev. B*, vol. 108, p. L020102, Jul 2023.
- [7] D. R. Hamann, "Optimized norm-conserving Vanderbilt pseudopotentials," *Phys. Rev. B*, vol. 88, p. 085117, Aug. 2013.
- [8] M. Schlipf and F. Gygi, "Optimization algorithm for the generation of ONCV pseudopotentials," *Comput. Phys. Commun.*, vol. 196, pp. 36–44, Nov. 2015.
- [9] A. Dal Corso, "Pseudopotentials periodic table: From H to Pu," *Computational Materials Science*, vol. 95, pp. 337–350, 2014.
- [10] X. Tao, A. Yang, S. Yang, Y. Quan, and P. Zhang, "Leading components and pressure-induced color changes in N-doped lutetium hydride," *Science Bulletin*, vol. 68, no. 13, pp. 1372–1378, 2023.
- [11] X. Liang, Z. Lin, J. Zhang, J. Zhao, S. Feng, W. Lu, G. Wang, L. Shi, N. Wang, P. Shan, Z. Zhang, M. Naamneh, R. Liu, B. Michon, J. Cheng, C. Jin, Y. Ren, and J. Ma, "Observation of flat band and van hove singularity in non-superconducting nitrogen-doped lutetium hydride," *arXiv preprint arXiv:2308.16420*, 2023.
- [12] P. Blaha, K. Schwarz, F. Tran, R. Laskowski, G. K. H. Madsen, and L. D. Marks, "WIEN2k: An APW+lo program for calculating the properties of solids," *The Journal of Chemical Physics*, vol. 152, p. 074101, 02 2020.

REVIEWERS' COMMENTS

Reviewer #1 (Remarks to the Author):

The authors have given detailed answers to every question raised and have revised the original manuscript accordingly. I recommend publishing this work.

Reviewer #2 (Remarks to the Author):

I have gone through the revised version and happy to see that authors have incorporated the changes suggested. I recommend the acceptance of the manuscript in its present form.